# AN EMPIRICAL STUDY OF EXAMPLE FORGETTING DURING DEEP NEURAL NETWORK LEARNING

**Mariya Toneva**[*†]
Carnegie Mellon University

**Alessandro Sordoni**[*]
Microsoft Research Montreal

**Remi Tachet des Combes**[*]
Microsoft Research Montreal

**Adam Trischler**
Microsoft Research Montreal

**Yoshua Bengio**
MILA, Université de Montréal
CIFAR Senior Fellow

**Geoffrey J. Gordon**
Microsoft Research Montreal
Carnegie Mellon University

## ABSTRACT

Inspired by the phenomenon of catastrophic forgetting, we investigate the learning dynamics of neural networks as they train on single classification tasks. Our goal is to understand whether a related phenomenon occurs when data does not undergo a clear distributional shift. We define a "forgetting event" to have occurred when an individual training example transitions from being classified correctly to incorrectly over the course of learning. Across several benchmark data sets, we find that: (i) certain examples are forgotten with high frequency, and some not at all; (ii) a data set's (un)forgettable examples generalize across neural architectures; and (iii) based on forgetting dynamics, a significant fraction of examples can be omitted from the training data set while still maintaining state-of-the-art generalization performance.

## 1 INTRODUCTION

Many machine learning models, in particular neural networks, cannot perform *continual learning*. They have a tendency to forget previously learnt information when trained on new tasks, a phenomenon usually called *catastrophic forgetting* (Kirkpatrick et al., 2017; Ritter et al., 2018). One of the hypothesized causes of catastrophic forgetting in neural networks is the shift in the input distribution across different tasks—*e.g.*, a lack of common factors or structure in the inputs of different tasks might lead standard optimization techniques to converge to radically different solutions each time a new task is presented. In this paper, we draw inspiration from this phenomenon and investigate the extent to which a related forgetting process occurs as a model learns examples traditionally considered to belong to *the same task*.

Similarly to the continual learning setting, in stochastic gradient descent (SGD) optimization, each mini-batch can be considered as a mini-"task" presented to the network sequentially. In this context, we are interested in characterizing the learning dynamics of neural networks by analyzing (catastrophic) *example forgetting events*. These occur when examples that have been "learnt" (*i.e.*, correctly classified) at some time $t$ in the optimization process are subsequently misclassified — or in other terms forgotten — at a time $t' > t$. We thus switch the focus from studying interactions between sequentially presented tasks to studying interactions between sequentially presented dataset examples during SGD optimization. Our starting point is to understand whether there exist examples that are consistently forgotten across subsequent training presentations and, conversely, examples that are never forgotten. We will call the latter *unforgettable* examples. We hypothesize that specific examples consistently forgotten between subsequent presentations, if they exist, must

---

[*]Equal contribution. Correspondence: MT: `mariya@cmu.edu`, AS: `alsordon@microsoft.com`
[†]Work done while interning at Microsoft Research Montreal

Code available at https://github.com/mtoneva/example_forgetting

not share commonalities with other examples from the same task. We therefore analyze the proportion of forgettable/unforgettable examples for a given task and what effects these examples have on a model's decision boundary and generalization error.

The goal of our investigation is two-fold. First, we attempt to gain insight into the optimization process by analyzing interactions among examples during learning and their influence on the final decision boundary. We are particularly interested in whether we can glean insight on the compressibility of a dataset, and thereby increase data efficiency without compromising generalization accuracy. It is a timely problem that has been the recent focus of few-shot learning approaches via meta-learning (Finn et al., 2017; Ravi & Larochelle, 2017). Second, we aim to characterize whether forgetting statistics can be used to identify "important" samples and detect *outliers* and examples with noisy labels (John, 1995; Brodley & Friedl, 1999; Sukhbaatar et al., 2014; Jiang et al., 2018).

Identifying important, or most informative examples is an important line of work and was extensively studied in the literature. Techniques of note — among others — are predefined curricula of examples (Bengio & LeCun, 2007), *self-paced* learning (Kumar et al., 2010), and more recently meta-learning (Fan et al., 2017). These research directions usually define "hardness" or "commonality" of an example as a function of the loss on that particular example at some point during training (or possibly at convergence). They do not consider whether some examples are consistently forgotten throughout learning. Very recently, Chang et al. (2017) consider re-weighting examples by accounting for the variance of their predictive distribution. This is related to our definition of forgetting events, but the authors provide little analysis of the extent to which the phenomenon occurs in their proposed tasks. Our purpose is to study this phenomenon from an empirical standpoint and characterize its prevalence in different datasets and across different model architectures.

Our experimental findings suggest that: a) there exist a large number of *unforgettable* examples, *i.e.*, examples that are never forgotten once learnt, those examples are stable across seeds and strongly correlated from one neural architecture to another; b) examples with noisy labels are among the most forgotten examples, along with images with "uncommon" features, visually complicated to classify; c) training a neural network on a dataset where a very large fraction of the least forgotten examples have been removed still results in extremely competitive performance on the test set.

## 2 RELATED WORK

**Curriculum Learning and Sample Weighting** Curriculum learning is a paradigm that favors learning along a curriculum of examples of increasing difficulty (Bengio et al., 2009). This general idea has found success in a variety of areas since its introduction (Kumar et al., 2010; Lee & Grauman, 2011; Schaul et al., 2015). Kumar et al. (2010) implemented their curriculum by considering easy the examples with a small loss. In our experiments, we empirically validate that unforgettable examples can be safely removed without compromising generalization. Zhao & Zhang (2015); Katharopoulos & Fleuret (2018) relate sample importance to the norm of its loss gradient with respect to the parameters of the network. Fan et al. (2017); Kim & Choi (2018); Jiang et al. (2018) learn a curriculum directly from data in order to minimize the task loss. Jiang et al. (2018) also study the robustness of their method in the context of noisy examples. This relates to a rich literature on outlier detection and removal of examples with noisy labels (John, 1995; Brodley & Friedl, 1999; Sukhbaatar et al., 2014; Jiang et al., 2018). We will provide evidence that noisy examples rank higher in terms of number of forgetting events. Koh & Liang (2017) borrow influence functions from robust statistics to evaluate the impact of the training examples on a model's predictions.

**Deep Generalization** The study of the generalization properties of deep neural networks when trained by stochastic gradient descent has been the focus of several recent publications (Zhang et al., 2016; Keskar et al., 2016; Chaudhari et al., 2016; Advani & Saxe, 2017). These studies suggest that the generalization error does not depend solely on the complexity of the hypothesis space. For instance, it has been demonstrated that over-parameterized models with many more parameters than training points can still achieve low test error (Huang et al., 2017; Wang et al., 2018) while being complex enough to fit a dataset with completely random labels (Zhang et al., 2016). A possible explanation for this phenomenon is a form of implicit regularization performed by stochastic gradient descent: deep neural networks trained with SGD have been recently shown to converge to the maximum margin solution in the linearly separable case (Soudry et al., 2017; Xu et al., 2018). In

our work, we provide empirical evidence that generalization can be maintained when removing a substantial portion of the training examples and without restricting the complexity of the hypothesis class. This goes along the support vector interpretation provided by Soudry et al. (2017).

## 3 DEFINING AND COMPUTING EXAMPLE FORGETTING

Our general case study for example forgetting is a standard classification setting. Given a dataset $\mathcal{D} = (\mathbf{x}_i, y_i)_i$ of observation/label pairs, we wish to learn the conditional probability distribution $p(y|\mathbf{x}; \theta)$ using a deep neural network with parameters $\theta$. The network is trained to minimize the empirical risk $R = \frac{1}{|\mathcal{D}|} \sum_i L(p(y_i|\mathbf{x}_i; \theta), y_i)$, where $L$ denotes the cross-entropy loss and $y_i \in 1, \ldots k$. The minimization is performed using variations of stochastic gradient descent, starting from initial random parameters $\theta^0$, and by sampling examples at random from the dataset $\mathcal{D}$.

**Forgetting and learning events** We denote by $\hat{y}_i^t = \arg\max_k p(y_{ik}|\mathbf{x}_i; \theta^t)$ the predicted label for example $\mathbf{x}_i$ obtained after $t$ steps of SGD. We also let $\mathrm{acc}_i^t = \mathbb{1}_{\hat{y}_i^t = y_i}$ be a binary variable indicating whether the example is correctly classified at time step $t$. Example $i$ undergoes a *forgetting event* when $\mathrm{acc}_i^t$ decreases between two consecutive updates: $\mathrm{acc}_i^t > \mathrm{acc}_i^{t+1}$. In other words, example $i$ is misclassified at step $t + 1$ after having been correctly classified at step $t$. Conversely, a *learning event* has occurred if $\mathrm{acc}_i^t < \mathrm{acc}_i^{t+1}$. Statistics that will be of interest in the next sections include the distribution of forgetting events across examples and the first time a learning event occurs.

**Classification margin** We will also be interested in analyzing the classification margin. Our predictors have the form $p(y_i|\mathbf{x}_i; \theta) = \sigma(\beta(\mathbf{x}_i))$, where $\sigma$ is a sigmoid (softmax) activation function in the case of binary (categorical) classification. The classification margin $m$ is defined as the difference between the logit of the correct class and the largest logit among the other classes, *i.e.* $m = \beta_k - \arg\max_{k' \neq k} \beta_{k'}$, where $k$ is the index corresponding to the correct class.

**Unforgettable examples** We qualify examples as *unforgettable* if they are learnt at some point and experience no forgetting events during the whole course of training: example $i$ is unforgettable if the first time it is learnt $t^*$ verifies $t^* < \infty$ and for all $k \geq t^*$, $\mathrm{acc}_i^k = 1$. Note that, according to this definition, examples that are never learnt during training do not qualify as unforgettable. We refer to examples that have been forgotten at least once as *forgettable*.

### 3.1 PROCEDURAL DESCRIPTION AND EXPERIMENTAL SETTING

Following the previous definitions, monitoring forgetting events entails computing the prediction for all examples in the dataset at each model update, which would be prohibitively expensive. In practice, for each example, we subsample the full sequence of forgetting events by computing forgetting statistics only when the example is included in the current mini-batch; that is, we compute forgetting across *presentations* of the same example in subsequent mini-batches. This gives a lower bound on the number of forgetting events an example undergoes during training.

We train a classifier on a given dataset and record the forgetting events for each example when they are sampled in the current mini-batch. For the purposes of further analysis, we then sort the dataset's examples based on the number of forgetting events they undergo. Ties are broken at random when sampling from the ordered data. Samples that are never learnt are considered forgotten an infinite number of times for sorting purposes. Note that this estimate of example forgetting is computationally expensive; see Sec. 6 for a discussion of a cheaper method.

---

**Algorithm 1** Computing forgetting statistics.

initialize $\mathrm{prev\_acc}_i = 0, i \in \mathcal{D}$
initialize forgetting $T[i] = 0, i \in \mathcal{D}$
**while** not training done **do**
    $B \sim \mathcal{D}$ # sample a minibatch
    **for** example $i \in B$ **do**
        compute $\mathrm{acc}_i$
        **if** $\mathrm{prev\_acc}_i > \mathrm{acc}_i$ **then**
            $T[i] = T[i] + 1$
        $\mathrm{prev\_acc}_i = \mathrm{acc}_i$
    gradient update classifier on $B$
**return** $T$

---

We perform our experimental evaluation on three datasets of increasing complexity: *MNIST* (LeCun et al., 1999), permuted MNIST – a version of *MNIST* that has the same fixed permutation applied to the pixels of all examples, and *CIFAR-10* (Krizhevsky, 2009). We use various model architectures and training schemes that yield test errors comparable

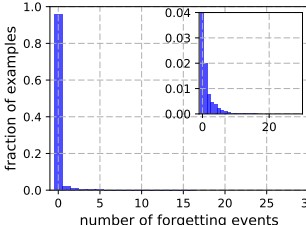 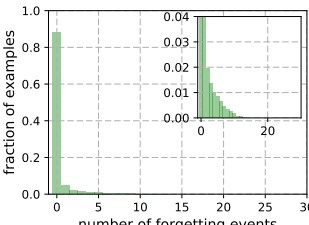 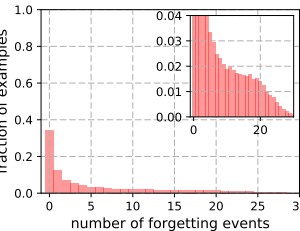

Figure 1: Histograms of forgetting events on (from left to right) *MNIST*, *permutedMNIST* and *CIFAR-10*. Insets show the zoomed-in y-axis.

with the current state-of-the-art on the respective datasets. In particular, the *MNIST*-based experiments use a network comprised of two convolutional layers followed by a fully connected one, trained using SGD with momentum and dropout. This network achieves $0.8\%$ test error. For *CIFAR-10*, we use a ResNet with cutout (DeVries & Taylor, 2017) trained using SGD and momentum with a particular learning rate schedule. This network achieves a competitive $3.99\%$ test error. For full experimentation details, see the Supplementary.

## 4   CHARACTERIZING EXAMPLE FORGETTING

**Number of forgetting events**   We estimate the number of forgetting events of all the training examples for the three different datasets (*MNIST*, *permutedMNIST* and *CIFAR-10*) across 5 random seeds. The histograms of forgetting events computed from one seed are shown in Figure 1. There are $55{,}012$, $45{,}181$ and $15{,}628$ unforgettable examples common across 5 seeds, they represent respectively $91.7\%$, $75.3\%$, and $31.3\%$ of the corresponding training sets. Note that datasets with less complexity and diversity of examples, such as *MNIST*, seem to contain significantly more unforgettable examples. *permutedMNIST* exhibits a complexity balanced between *MNIST* (easiest) and *CIFAR-10* (hardest). This finding seems to suggest a correlation between forgetting statistics and the intrinsic dimension of the learning problem, as recently formalized by Li et al. (2018).

**Stability across seeds** To test the stability of our metric with respect to the variance generated by stochastic gradient descent, we compute the number of forgetting events per example for $10$ different random seeds and measure their correlation. From one seed to another, the average Pearson correlation is $89.2\%$. When randomly splitting the $10$ different seeds into two sets of 5, the cumulated number of forgetting events within those two sets shows a high correlation of $97.6\%$. We also ran the original experiment on 100 seeds to devise $95\%$ confidence bounds on the average (over 5 seeds) number of forgetting events per example (see Appendix 13). The confidence interval of the least forgotten examples is tight, confirming that examples with a small number of forgetting events can be ranked confidently.

**Forgetting by chance** In order to quantify the possibility of forgetting occurring by chance, we additionally analyze the distribution of forgetting events obtained under the regime of random update steps instead of the true SGD steps. In order to maintain the statistics of the random updates similar to those encountered during SGD, random updates are obtained by shuffling the gradients produced by standard SGD on a main network (more details are provided in Appendix 12). We report the histogram of chance forgetting events in Supplementary Figure 13: examples are being forgotten by chance a small number of time, at most twice and most of the time less than once. The observed stability across seeds, low number of chance forgetting events and the tight confidence bounds suggest that it is unlikely for the ordering produced by the metric to be the by-product of another unrelated random cause.

**First learning events**   We investigate whether unforgettable and forgettable examples need to be presented different numbers of times in order to be learnt for the first time (*i.e.* for the first learning event to occur, as defined in Section 3). The distributions of the presentation numbers at which first learning events occur across all datasets can be seen in Supplemental Figure 8. We observe that, while both unforgettable and forgettable sets contain many examples that are learnt during the

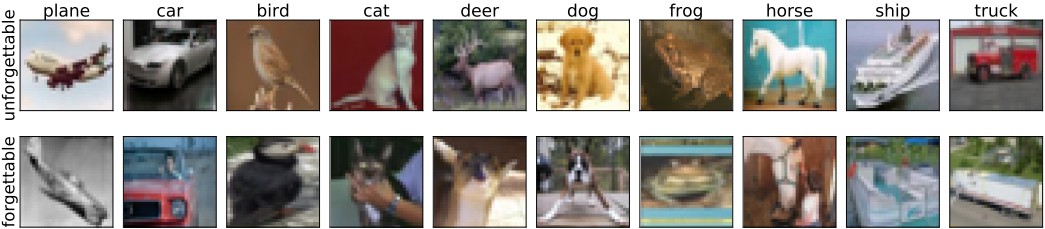

Figure 2: Pictures of unforgettable (*Top*) and forgettable examples (*Bottom*) of every *CIFAR-10* class. Forgettable examples seem to exhibit peculiar or uncommon features. Additional examples are available in Supplemental Figure 15.

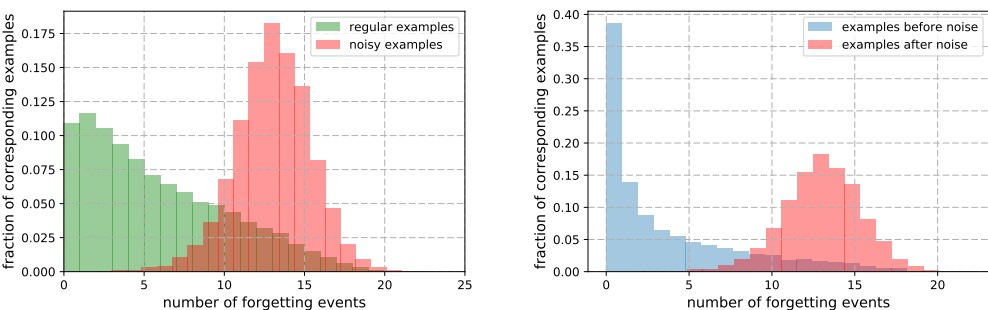

Figure 3: Distributions of forgetting events across training examples in *CIFAR-10* when $20\%$ of labels are randomly changed. *Left.* Comparison of forgetting events between examples with noisy and original labels. The most forgotten examples are those with noisy labels. No noisy examples are unforgettable. *Right.* Comparison of forgetting events between examples with noisy labels and the same examples with original labels. Examples exhibit more forgetting when their labels are changed.

first 3-4 presentations, the forgettable examples contain a larger number of examples that are first learnt later in training. The Spearman rank correlation between the first learning event presentations and the number of forgetting events across all training examples is $0.56$, indicating a moderate relationship.

**Misclassification margin** The definition of forgetting events is binary and as such fairly crude compared to more sophisticated estimators of example relevance (Zhao & Zhang, 2015; Chang et al., 2017). In order to qualify its validity, we compute the misclassification margin of forgetting events. The misclassification margin of an example is defined as the mean classification margin (defined in Section 3) over all its forgetting events, a negative quantity by definition. The Spearman rank correlation between an example's number of forgetting events and its mean misclassification margin is -0.74 (computed over 5 seeds, see corresponding 2D-histogram in Supplemental Figure 9). These results suggest that examples which are frequently forgotten have a large misclassification margin.

**Visual inspection** We visualize some of the unforgettable examples in Figure 2 along with some examples that have been most forgotten in the *CIFAR-10* dataset. Unforgettable samples are easily recognizable and contain the most obvious class attributes or centered objects, *e.g.*, a plane on a clear sky. On the other hand, the most forgotten examples exhibit more ambiguous characteristics (as in the center image, a truck on a brown background) that may not align with the learning signal common to other examples from the same class.

**Detection of noisy examples** We further investigate the observation that the most forgettable examples seem to exhibit atypical characteristics. We would expect that if highly forgettable examples have atypical class characteristics, then noisily-labeled examples will undergo more forgetting

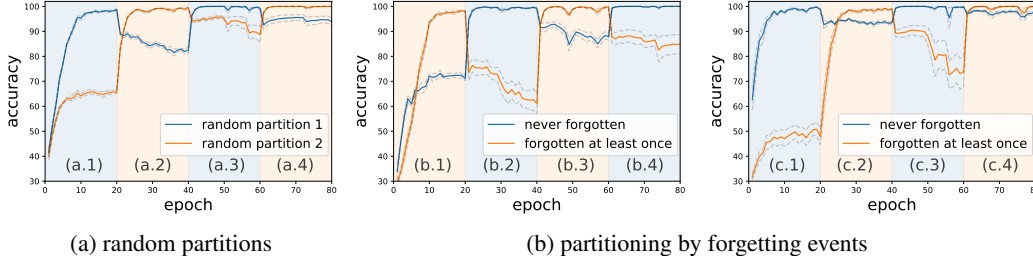

(a) random partitions                       (b) partitioning by forgetting events

Figure 4: Synthetic continual learning setup for *CIFAR-10*. Background color in each column indicates the training partition, curves track performance on both partitions during interleaved training. Solids lines represent the average of 5 runs and dashed lines represent the standard error. The figure highlights that examples that have been forgotten at least once can "support" those that have never been forgotten, as shown in (c.2) and (b.3).

events. We randomly change the labels of $20\%$ of *CIFAR-10* and record the number of forgetting events of both the noisy and regular examples through training. The distributions of forgetting events across noisy and regular examples are shown in Figure 3. We observe that the most forgotten examples are those with noisy labels and that no noisy examples are unforgettable. We also compare the forgetting events of the noisy examples to that of the same set of examples with original labels and observe a much higher degree of forgetting in the noisy case. The results of these synthetic experiments support the hypothesis that highly forgettable examples exhibit atypical class characteristics.

## 4.1 CONTINUAL LEARNING SETUP

We observed that in harder tasks such as *CIFAR-10*, a significant portion of examples are forgotten at least once during learning. This leads us to believe that catastrophic forgetting may be observed, to some extent, even when considering examples coming from the same task distribution. To test this hypothesis, we perform an experiment inspired by the standard continual learning setup (McCloskey & Cohen, 1989; Kirkpatrick et al., 2017). We create two tasks by randomly sampling 10k examples from the *CIFAR-10* training set and dividing them in two equally-sized partitions (5k examples each). We treat each partition as a separate "task" even though they should follow the same distribution. We then train a classifier for 20 epochs on each partition in an alternating fashion, while tracking performance on both partitions. The results are reported in Figure 4 (a). The background color represents which of the two partitions is currently used for training. We observe some forgetting of the second task when we only train on the first task (panel (a.2)). This is somewhat surprising as the two tasks contain examples from the same underlying distribution.

We contrast the results from training on random partitions of examples with ones obtained by partitioning the examples based on forgetting statistics (Figure 4 (b)). That is, we first compute the forgetting events for all examples based on Algorithm 1 and we create our tasks by sampling 5k examples that have zero forgetting events (named f0) and 5k examples that have non-zero forgetting events (named fN). We observe that examples that have been forgotten at least once suffer a more drastic form of forgetting than those included in a random split (compare (a.2) with (b.2)). In panel (b.3) and (c.2) we can observe that examples from task f0 suffer very mild forgetting when training on task fN. This suggests that examples that have been forgotten at least once may be able to "support" those that have never been forgotten. We observe the same pattern when we investigate the opposite alternating sequence of tasks in Figure 4 (b, right).

## 5 REMOVING UNFORGETTABLE EXAMPLES

As shown in the previous section, learning on examples that have been forgotten at least once minimally impacts performance on those that are unforgettable. This appears to indicate that unforgettable examples are less informative than others, and, more generally, that the more an example is forgotten during training, the more useful it may be to the classification task. This seems to align with the observations in Chang et al. (2017), where the authors re-weight training examples by ac-

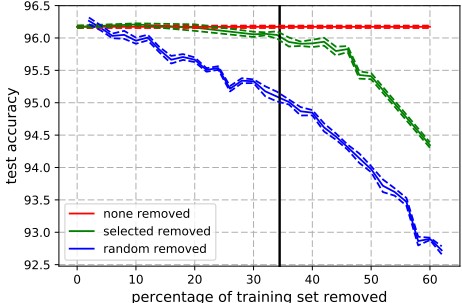 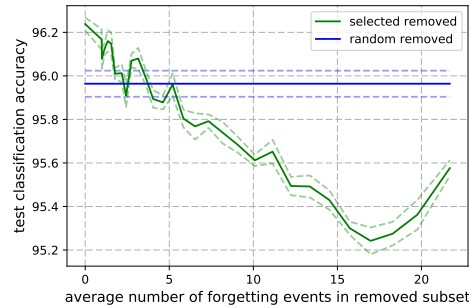

Figure 5: *Left* Generalization performance on *CIFAR-10* of ResNet18 where increasingly larger subsets of the training set are removed (mean +/- std error of 5 seeds). When the removed examples are selected at random, performance drops very fast. Selecting the examples according to our ordering can reduce the training set significantly without affecting generalization. The vertical line indicates the point at which all unforgettable examples are removed from the training set. *Right* Difference in generalization performance when contiguous chunks of 5000 increasingly forgotten examples are removed from the training set. Most important examples tend to be those that are forgotten the most.

counting for the variance of their predictive distribution. Here, we test whether it is possible to completely remove a given subset of examples during training.

In Fig. 5 (*Left*), we show the evolution of the generalization performance in *CIFAR-10* when we artificially remove examples from the training dataset. We choose the examples to remove by increasing number of forgetting events. Each point in the figure corresponds to retraining the model from scratch on an increasingly smaller subset of the training data (with the same hyper-parameters as the base model). We observe that when removing a random subset of the dataset, performance rapidly decreases. Comparatively, by removing examples ordered by number of forgetting events, 30% of the dataset can be removed while maintaining comparable generalization performance as the base model trained on the full dataset, and up to 35% can be removed with marginal degradation (less than 0.2%). The results on the other datasets are similar: a large fraction of training examples can be ignored without hurting the final generalization performance of the classifiers (Figure 6).

In Figure 5 (*Right*), we show the evolution of the generalization error when we remove from the dataset 5,000 examples with increasing forgetting statistics. Each point in the figure corresponds to the generalization error of a model trained on the full dataset minus 5,000 examples as a function of the average number of forgetting events in those 5,000 examples. As can be seen, removing the same number of examples with increasingly more forgetting events results in worse generalization for most of the curve. It is interesting to notice the rightmost part of the curve moving up, suggesting that some of the most forgotten examples actually hurt performance. Those could correspond to outliers or mislabeled examples (see Sec. 4). Finding a way to separate those points from very informative ones is an ancient but still active area of research (John, 1995; Jiang et al., 2018).

**Support vectors** Various explanations of the *implicit generalization* of deep neural networks (Zhang et al., 2016) have been offered: flat minima generalize better and stochastic gradient descent converges towards them (Hochreiter & Schmidhuber, 1997; Kleinberg et al., 2018), gradient descent protects against overfitting (Advani & Saxe, 2017; Tachet et al., 2018), deep networks' structure biases learning towards simple functions (Neyshabur et al., 2014; Perez et al., 2018). But it remains a poorly understood phenomenon. An interesting direction of research is to study the convergence properties of gradient descent in terms of maximum margin classifiers. It has been shown recently (Soudry et al., 2017) that on separable data, a linear network will learn such a maximum margin classifier. This supports the idea that stochastic gradient descent implicitly converges to solutions that maximally separate the dataset, and additionally, that some data points are more relevant than others to the decision boundary learnt by the classifier. Those points play a part equivalent to support vectors in the *support vector machine* paradigm. Our results confirm that a significant portion of training data points have little to no influence on the generalization performance when the decision function is learnt with SGD. Forgettable training points may be considered as analogs to support vec-

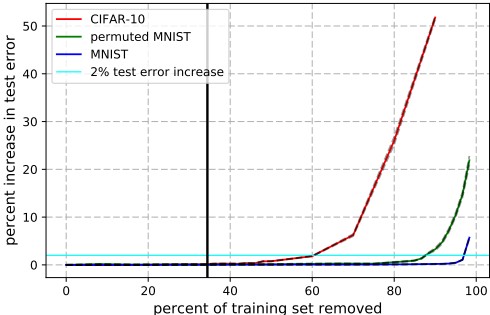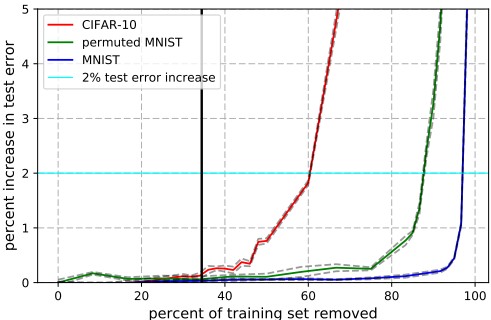

Figure 6: Decrease in generalization performance when fractions of the training sets are removed. When the subsets are selected appropriately, performance is maintained after removing up to 30% of *CIFAR-10*, 50% of *permutedMNIST*, and 80% of *MNIST*. Vertical black line indicates the point at which all unforgettable examples are removed from CIFAR-10. *Right* is a zoomed in version of *Left*.

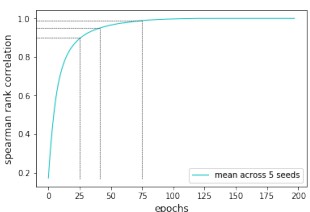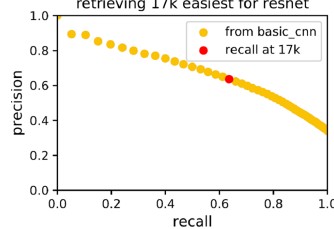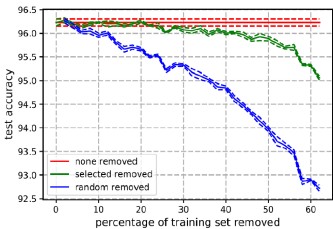

Figure 7: *Left.* Ranking of examples by forgotten events stabilizes after 75 epochs in *CIFAR-10*. *Middle.* Precision and recall of retrieving the unforgettable examples of ResNet18, using the example ordering of a simpler convolutional neural network. *Right.* Generalization performance on *CIFAR-10* of a WideResNet using the example ordering of ResNet18.

tors, important for the generalization performance of the model. The number of forgetting events of an example is a relevant metric to detect such support vectors. It also correlates well with the misclassification margin (see Sec.4) which is a proxy for the distance to the decision boundary.

**Intrinsic dataset dimension** As mentioned above, the datasets we study have various fractions of unforgettable events (91.7% for *MNIST*, 75.3% for *permutedMNIST* and 31.3% for *CIFAR-10*). We also see in Figure 6 that performance on those datasets starts to degrade at different fractions of removed examples: the number of support vectors varies from one dataset to the other, based on the complexity of the underlying data distribution. If we assume that we are in fact detecting analogs of support vectors, we can put these results in perspective with the intrinsic dataset dimension defined by Li et al. (2018) as the codimension in the parameter space of the solution set: for a given architecture, the higher the intrinsic dataset dimension, the larger the number of support vectors, and the fewer the number of unforgettable examples.

## 6 TRANSFERABLE FORGETTING EVENTS

Forgetting events rely on training a given architecture, with a given optimizer, for a given number of epochs. We investigate to what extent the forgetting statistics of examples depend on those factors.

**Throughout training** We compute the Spearman rank correlation between the ordering obtained at the end of training (200 epochs) and the ordering after various number of epochs. As seen in Fig. 7 (*Left*), the ordering is very stable after 75 epochs, and we found a reasonable number of epochs to get a good correlation to be 25 (see the Supplementary Materials for precision-recall plots).

**Between architectures** A limitation of our method is that it requires computing the ordering from a previous run. An interesting question is whether that ordering could be obtained from a simpler architecture than residual networks. We train a network with two convolutional layers followed by two fully connected ones (see the Supplementary for the full architecture) and compare the resulting ordering with the one obtained with ResNet18. Figure 7 (*Middle*) shows a precision-recall plot of the unforgettable examples computed with the residual network. We see a reasonably strong agreement between the unforgettable examples of the convolutional neural network and the ones of the ResNet18. Finally, we train a WideResNet (Zagoruyko & Komodakis, 2016) on truncated data sets using the example ordering from ResNet18. Using the same computing power (one Titan X GPU), Resnet18 requires 2 hours to train whereas WideResNet requires 8 – estimating the forgetting statistics of WideResNet via ResNet18 can save up to 6 hours of training time if the estimate is accurate. We plot WideResNet's generalization performance using the ordering obtained by ResNet18 in Figure 7 (*Right*): the network still performs near optimally with 30% of the dataset removed. This opens up promising avenues of computing forgetting statistics with smaller architectures.

## 7 CONCLUSION AND FUTURE WORK

In this paper, inspired by the phenomenon of catastrophic forgetting, we investigate the learning dynamics of neural networks when training on single classification tasks. We show that catastrophic forgetting can occur in the context of what is usually considered to be a single task. Inspired by this result, we find that some examples within a task are more prone to being forgotten, while others are consistently unforgettable. We also find that forgetting statistics seem to be fairly stable with respect to the various characteristics of training, suggesting that they actually uncover intrinsic properties of the data rather than idiosyncrasies of the training schemes. Furthermore, the unforgettable examples seem to play little part in the final performance of the classifier as they can be removed from the training set without hurting generalization. This supports recent research interpreting deep neural networks as max margin classifiers in the linear case. Future work involves understanding forgetting events better from a theoretical perspective, exploring potential applications to other areas of supervised learning, such as speech or text and to reinforcement learning where forgetting is prevalent due to the continual shift of the underlying distribution.

## 8 ACKNOWLEDGMENTS

We acknowledge the anonymous reviewers for their insightful suggestions.

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

# 9 EXPERIMENTATION DETAILS

***Detailed distributions***

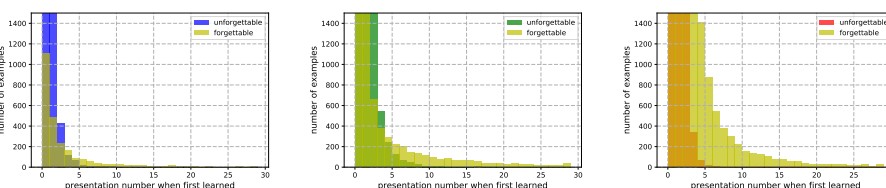

Figure 8: From left to right, distributions of the first presentation at which each unforgettable and forgettable example was learned in *MNIST*, *permutedMNIST* and *CIFAR-10* respectively. Rescaled view where the number of examples have been capped between 0 and 1500 for visualization purposes. Unforgettable examples are generally learnt early during training, thus may be considered as "easy" in the sense of Kumar et al. (2010), i.e. may have a low loss during most of the training.

***Misclassification margin***

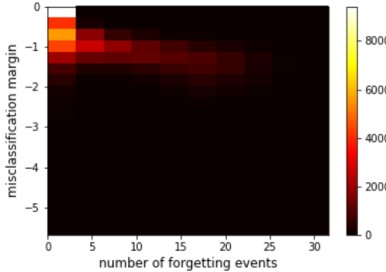

Figure 9: *Left* 2D-histogram of the number of forgetting events and mean misclassification margin across all examples of *CIFAR-10*. There is significant negative correlation (-0.74, Spearman rank correlation) between mean misclassification margin and the number of forgetting events.

***permutedMNIST*** The *permutedMNIST* data set is obtained by applying a fixed random permutation of the pixels to all the images of the standard *MNIST* data set. This typically makes the data set harder to learn for convolutional neural networks as local patterns, *e.g.* the horizontal bar of the 7, get shuffled. This statement is supported by the two following facts:

- The number of unforgettable examples for *permutedMNIST* is 45181 versus 55012 for *MNIST*.
- The intrinsic data set dimension (Li et al., 2018) of *permutedMNIST* is 1400 compared to 290 for the untouched data set.

**Network Architectures** We use a variety of different architectures in the main text. Below are their specifications.

The architecture for the *MNIST* and *permutedMNIST* experiments is the following:

1. a first convolutional layer with 5 by 5 filters and 10 feature maps,
2. a second convolutional layer with 5 by 5 filters and 20 feature maps,
3. a fully connected layer with 50 hidden units
4. the output layer, with 10 logits, one for each class.

We apply ReLU nonlinearities to the feature maps and to the hidden layer. The last layer is passed through a softmax to output probabilities for each class of the data set.

The ResNet18 architecture used for *CIFAR-10* is described thoroughly in DeVries & Taylor (2017), its implementation can be found at `https://github.com/uoguelph-mlrg/Cutout`.

The second one is a WideResNet (Zagoruyko & Komodakis, 2016), with a depth of 28 and a widen factor of 10. We used the implementation found at `https://github.com/meliketoy/wide-resnet.pytorch`.

The convolutional architecture used in Section 6 is the following:

1. a first convolutional layer with 5 by 5 filters and 6 feature maps,
2. a 2 by 2 max pooling layer
3. a second convolutional layer with 5 by 5 filters and 16 feature maps,
4. a first fully connected layer with 120 hidden units
5. a second fully connected layer with 84 hidden units
6. the output layer, with 10 logits, one for each class.

**Optimization**

The *MNIST* networks are trained to minimize the cross-entropy loss using stochastic gradient descent with a learning rate of $0.01$ and a momentum of $0.5$.

The ResNet18 is trained using cutout, data augmentation and stochastic gradient descent with a $0.9$ Nesterov momentum and a learning rate starting at $0.1$ and divided by 5 at epochs $60$, $120$ and $160$.

The WideResNet is trained using Adam (Kingma & Ba, 2014) and a learning rate of $0.001$.

## 10 STABILITY OF THE FORGETTING EVENTS

In Fig 10, we plot precision-recall diagrams for the unforgettable and most forgotten examples of *CIFAR-10* obtained on ResNet18 after 200 epochs and various prior time steps. We see in particular that at 75 epochs, the examples on both side of the spectrum can be retrieved with very high precision and recall.

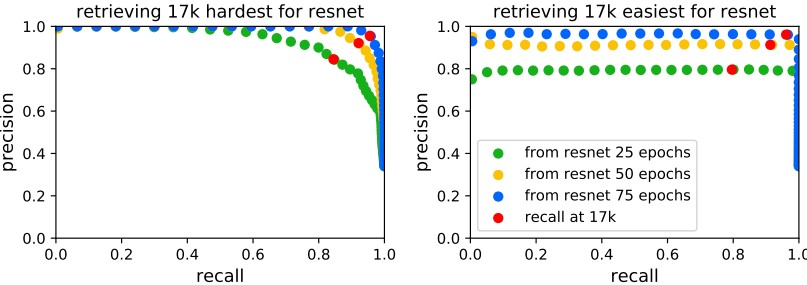

Figure 10: Right: precision and recall of retrieving the unforgettable examples from a full run of ResNet18 (200 epochs), using the example ordering after 25, 50, and 75 epochs. The unforgettable examples are retrieved with high precision and recall after 50 epochs. Left: same plot for the 17k examples with the most forgetting events.

## 11 *Noising* THE DATA SETS

In Section 4, we analyzed the effect of adding *label noise* on the distribution of forgetting events. Here, we examine the effect of adding *pixel noise*, i.e. noising the input distribution. We choose to corrupt the inputs with additive Gaussian noise with zero mean and we choose for its standard deviation to be a multiple of channel-wise data standard deviation (i.e., $\sigma_{\text{noise}} = \lambda \sigma_{\text{data}}, \lambda \in \{0.5, 1, 2, 10\}$). Note that we add the noise after applying a channel-wise standard normalization

step of the training images, therefore $\sigma_{\text{data}} = 1$ (each channel has zero mean, unit variance, this is a standard pre-processing step and has been applied throughout all the experiments in this paper).

The forgetting distributions obtained by noising all the dataset examples with increasing noise standard deviation are presented in Figure 11. We observe that adding increasing amount of noise decreases the amount of unforgettable examples and increases the amount of examples in the second mode of the forgetting distribution.

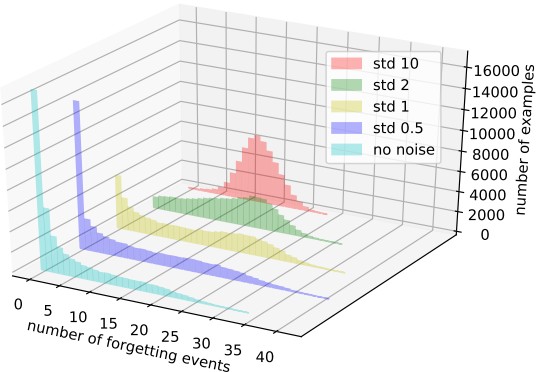

Figure 11: Distribution of forgetting events across all training examples in *CIFAR-10* when all training images are augmented with increasing additive Gaussian noise. The presence of increasing amount of noise decreases the amount of unforgettable examples and increases the amount of examples in the second mode of the forgetting distribution.

We follow the noisy-labels experiments of Section 4 and we apply the aforementioned pixel noise to 20% of the training data ($\sigma_{\text{noise}} = 10$). We present the results of comparing the forgetting distribution of the 20% of examples before and after noise was added to the pixels in Figure 12 (Left). For ease of comparison, we report the same results in the case of label noise in Figure 12 (Right). We observe that the forgetting distribution under pixel noise resembles the one under label noise.

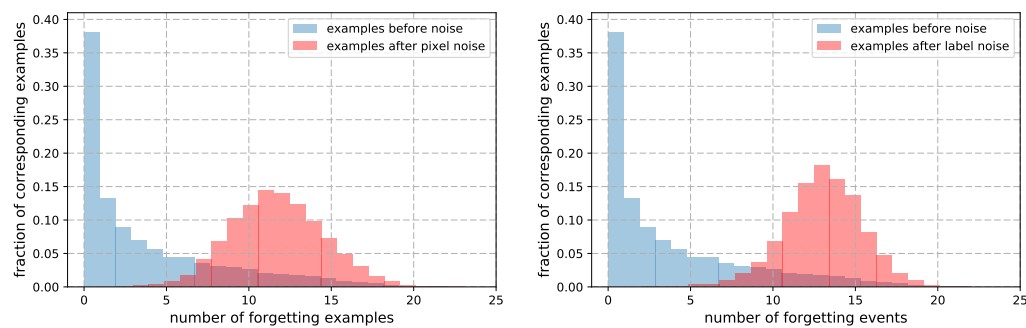

Figure 12: Distribution of forgetting events across all training examples in *CIFAR-10* when random 20% of training examples undergo *pixel noise* ($\sigma_{\text{noise}} = 10$) *(Left)* or *label noise (Right)* (same as Figure 3). We observe that the forgetting distribution under pixel noise resembles the one under label noise.

## 12 "CHANCE" FORGETTING EVENTS ON *CIFAR-10*

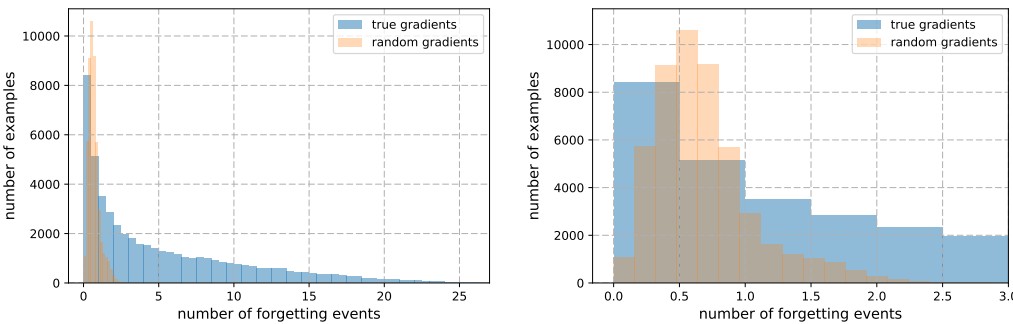

Figure 13: Histogram of forgetting events under true and random gradient steps. (*Right*) Zoomed-in version where the number of forgetting events is capped at 3 for visualization.

Forgetting events may happen by "chance", i.e. some learning/forgetting events may occur even with random gradients. In order to estimate how large the effect of "chance" is, we compute the forgetting events of a classifier obtained by randomizing the update steps. To keep the statistics of the gradients similar to those encountered during SGD, we proceed as follows:

1. Before the beginning of training, clone the "base" classifier into a new "clone" classifier with the same random weights.

2. At each training step, shuffle the gradients computed on the base classifier and apply those to the clone (the base classifier is still optimized the same way): this ensures that the statistics of the random updates match the statistics of the true gradients during learning.

3. Compute the forgetting events of the clone classifier on the training set exactly as is done with the base classifier.

The results can be found in Fig 13, showing the histogram of forgetting events produced by the clone network, averaged over 5 seeds. This gives an idea of the chance forgetting rate across examples. In this setting, examples are being forgotten by chance at most twice.

## 13 CONFIDENCE ON FORGETTING EVENTS FOR *CIFAR-10*

In order to establish confidence intervals on the number of forgetting events, we computed them on 100 seeds and formed 20 averages over 5 seeds. In Fig 14, we show the average (in green), the bottom 2.5 percentile (in blue) and top 2.5 percentile (in orange) of those 20 curves.

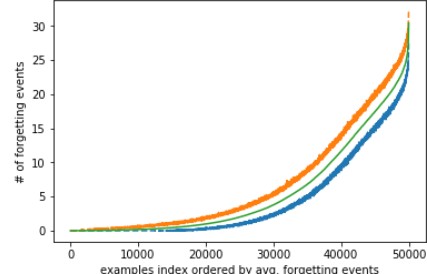

Figure 14: 95% confidence interval on forgetting events averaged over 5 seeds.

## 14 VISUALIZATION OF FORGETTABLE AND UNFORGETTABLE IMAGES

See Fig 15 for additional pictures of the most unforgettable and forgettable examples of every *CIFAR-10* class, when examples are sorted by number of forgetting events (ties are broken randomly).

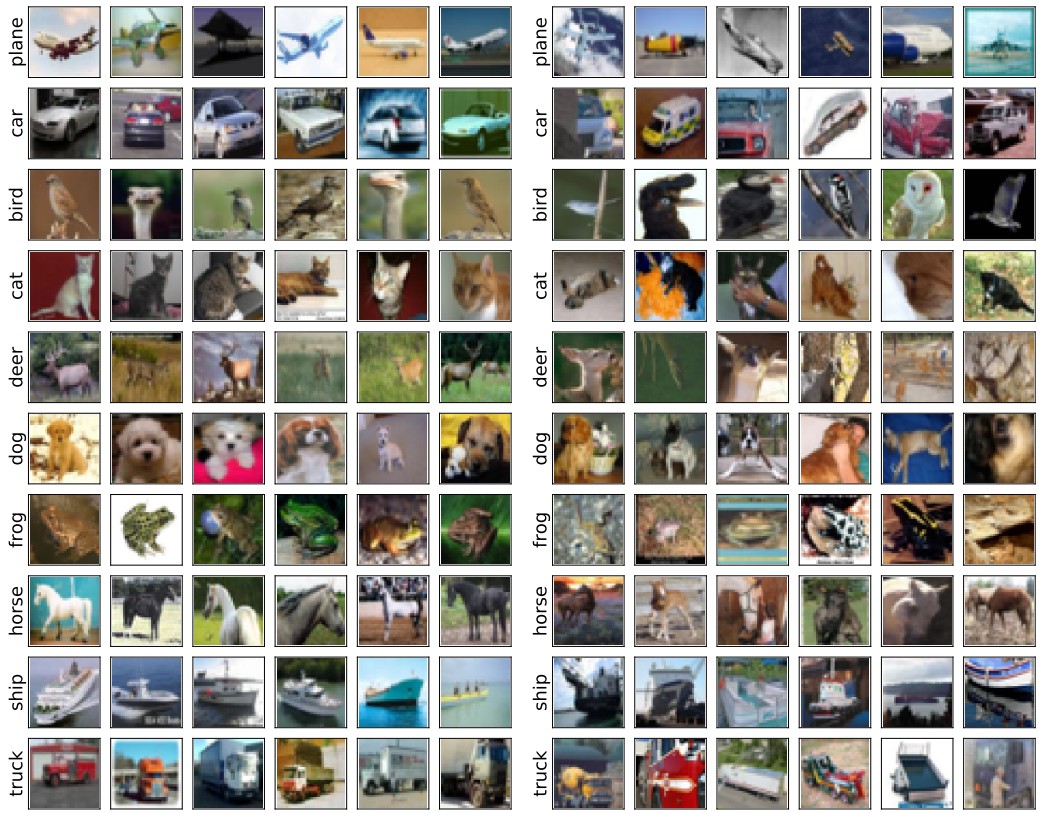

Figure 15: Additional pictures of the most unforgettable (*Left*) and forgettable examples (*Right*) of every *CIFAR-10* class, when examples are sorted by number of forgetting events (ties are broken randomly). Forgettable examples seem to exhibit peculiar or uncommon features.

## 15 FORGETTING IN *CIFAR-100*

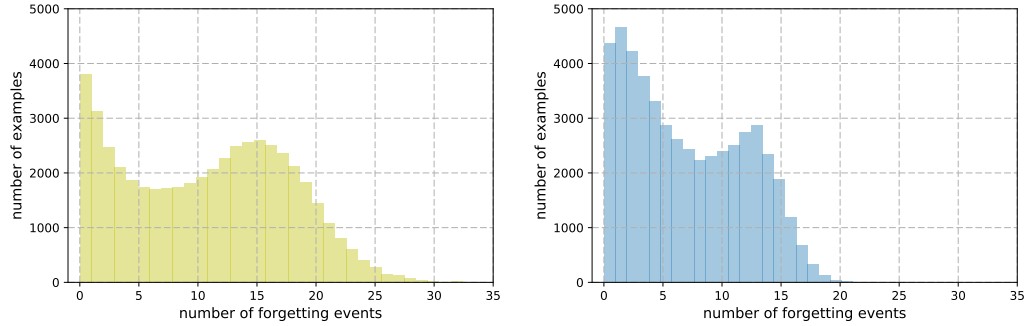

Figure 16: Left: distribution of forgetting events in *CIFAR-100*. Right: distribution of forgetting events in *CIFAR-10* when 20% of the labels are changed at random. The distribution of forgetting in *CIFAR-100* is much closer to that of forgetting in the noisy *CIFAR-10* than it is to forgetting in the original datasets presented in Figure 1.

The distribution of forgetting events in *CIFAR-100* is shown in Figure 16. There are 3809 unforgettable examples (7.62% of the training set). *CIFAR-100* is the hardest to classify out all of the presented datasets and exhibits the highest percentage of forgetting events. This finding further supports the idea that there may be a correlation between the forgetting statistics and the intrinsic

dimension of the learning problem. Additionally, each *CIFAR-100* class contains 10 times fewer examples than in *CIFAR-10* or the *MNIST* datasets, making each image all the more useful for the learning problem.

We also observe that the distribution of forgetting in *CIFAR-100* is much closer to that of forgetting in the noisy *CIFAR-10* than it is to forgetting in the original datasets presented in Figure 1. Visualizing the most forgotten examples in *CIFAR-100* revealed that *CIFAR-100* contains several images that appear multiple times in the training set under different labels. In Figure 17, we present the 36 most forgotten examples in *CIFAR-100*. Note that they are all images that appear under multiple labels (not shown: the "girl" image also appears under the label "baby", the "mouse" image also appears under "shrew", one of the 2 images of 'oak_tree' appears under 'willow_tree' and the other under 'maple_tree').

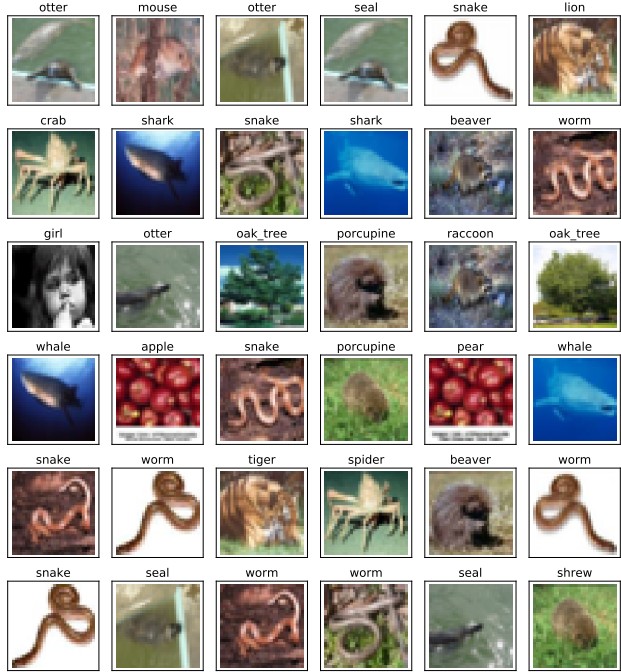

Figure 17: The 36 most forgotten examples in *CIFAR-100*. Note that they are all images that appear under multiple labels (not pictured: the "girl" image also appears under the label "baby", the "mouse" image also appears under "shrew", one of the 2 images of 'oak_tree' appears under 'willow_tree' and the other under 'maple_tree'.

We perform the same removal experiments we presented in Figure 5 for *CIFAR-100*. The results are shown in Figure 18. Just like with *CIFAR-10*, we are able to remove all unforgettable examples ( $8\%$ of the training set) while maintaining test performance.

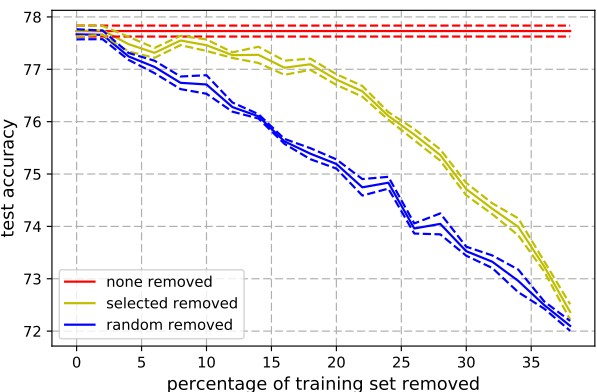

Figure 18: Generalization performance on *CIFAR-100* of ResNet18 where increasingly larger subsets of the training set are removed (mean +/- std error of 5 seeds). When the removed examples are selected at random, performance drops faster. Selecting the examples according to our ordering reduces the training set without affecting generalization.

