# OpenReview forum: "An Empirical Study of Example Forgetting during Deep Neural Network Learning"
_ICLR.cc/2019/Conference_

### Official Review · AnonReviewer2 · 2018-10-23
**Thorough experiments which prove there exist "support examples" in neural network training.**

**Rating:** 7
**Confidence:** 4

**Review:**

This paper studies the forgetting behavior of the training examples during SGD. Empirically it shows there are forgettable and unforgettable examples, unforgettable examples are like "support examples", one can achieve similar performance by training only on these "support examples". The paper also shows this phenomenon is consistent across different network architectures.

Pros:
This paper is written in high quality, clearly presented. It is original in the sense that this is the first empirical study on the forgettability of examples in during neural network training.

Comments and Questions on the experiment details:
1. Is the dataset randomly shuffled after every epoch? One concern is that if the order is fixed, some of the examples will be unforgettable simply because the previous batches have similar examples , and training the model on the previous batches makes it good on some examples in the current batch.
2. It would be more interesting to also include datasets like cifar100, which has more labels. The current datasets all have only 10 categories.
3. An addition figure can be provided which switches the order of training in figure 4b. Namely, start with training on b.2.

Cons:
Lack of insight. Subjectively, I usually expect empirical analysis papers to either come up with unexpected observations or provide guidance for practice. In my opinion, the findings of this work is within expectation, and there is a gap for practice.

Overall this paper is worth publishing for the systematic experiments which empirically verifies that there are support examples in neural networks.

---

> ### Author Response · Authors · 2018-11-14
> **Clarifications and Additional Investigation on CIFAR-100**
>
> Thanks for your interesting review. We try to address your remarks below:
>
> 1) We randomly shuffle all datasets at the start of each epoch.
>
> 2) As suggested, we investigated forgetting in CIFAR-100. We show the detailed results in Appendix 14 of the updated paper. In short, we observe that about 8% of examples in CIFAR-100 are unforgettable, which is the lowest percentage out of all investigated datasets: CIFAR-100 contains 10 times fewer examples per class (500 examples per class) than CIFAR-10 or the MNIST datasets, making each image all the more useful for the learning problem.
>
> Unexpectedly, we observed that the distribution of forgetting events in CIFAR-100 resembles the distribution of forgetting events in the noisy CIFAR-10 (with 20% randomly changed labels). This led us to suspect that a portion of CIFAR-100 examples could have noisy labels. Upon visualization of the most forgotten examples in CIFAR-100, we discovered that there are several images that appear under multiple labels, introducing noise to the dataset and possibly diminishing the proportion of unforgettable examples.
>
> For completeness, we added the removal experiments from Figure 5 (Left) for CIFAR-100 to Appendix 14. The results align with those from the other datasets -- we are able to remove all unforgettable examples and maintain generalization performance, while outperforming a random removal baseline.
>
> 3) We have included the experiment in the main paper in Figure 4 (right). Note that the "never forgotten" set continues to suffer from less degradation when training on the "forgotten at least once" set.

---

> > ### Comment · AnonReviewer2 · 2018-11-19
> > **Response to Author**
> >
> > Thank you for the clarification and extra experiments on CIFAR-100.
> > Overall, this is a paper with high quality, the experiments are complete and the paper is well written. I'm increasing the score to 7.
> > I'm not giving a higher score because I think the impact of this paper on solving the catastrophic forgetting problem seems limited.

---

> > > ### Author Response · Authors · 2018-11-21
> > > **Alleviating Catastrophic Forgetting as One of Several Exciting Future Steps**
> > >
> > > Thanks for your review and suggestions, your suggested additional experiments have strengthened the paper and we will acknowledge them in the paper, if accepted. Applying some of our results towards solving catastrophic forgetting is one of the promising directions we hope to investigate in the future. One of the paths we are currently investigating is whether we can build focused memories of representative examples from previous tasks. Nonetheless, we believe our current analysis to be general, and, as such we keep the hope that our results could potentially be helpful in an even larger set of problems.

---

### Official Review · AnonReviewer1 · 2018-11-02
**Review of "An Empirical Study of Example Forgetting during Deep Neural Network Learning"**

**Rating:** 8
**Confidence:** 4

**Review:**

UPDATE 2 (Nov 19, 2018): The paper has improved very substantially since the initial submission, and the authors have addressed almost all of my comments. I have therefore increased my score to an 8 and recommend acceptance.
------------------------------------------------------------------------------------------------------------------------------

UPDATE (Nov 16, 2018) : In light of the author response, I have increased my score to a 6.
------------------------------------------------------------------------------------------------------------------------------

This paper aims to analyze the extent to which networks learn to correctly classify specific examples and then “forget” these examples over the course of training. The authors provide several examples of forgettable and unforgettable examples, demonstrating, among other things, that examples with noisy examples are more forgettable and that a reasonable fraction of unforgettable examples can be removed from the training set without harming performance.

The paper is clearly written, and the work is novel -- to my knowledge, this is the first investigation of example forgetting over training. There are an interesting and likely important set of ideas here, and portions of the paper are quite strong -- in particular, the experiment demonstrating that examples with noisy examples are more forgettable is quite nice. However, there are several experimental oversights which make this paper difficult to recommend for publication in its current form.

Major points:

1) The most critical issue is with the measurement of forgetting itself: the authors do not take into account the chance forgetting rate in any of their experiments. Simply due to chance, some examples will be correctly labeled at some point in training (especially in the datasets analyzed, which only contain 10 classes). This makes it difficult to distinguish whether a “forgotten” example was actually ever learned in the first place. In order to properly ground this metric, measurements of chance forgetting rates will be necessary (for example, what are the forgetting rates when random steps are taken at each update step?).

2) Were the networks trained on MNIST, permutedMNIST, and CIFAR-10 trained for the same number of epochs? Related to point 1, the forgetting rate should increase with the number of epochs used in training as the probability of each example being correctly classified should increase. If the CIFAR-10 models were trained for more epochs, this would explain the observation that more CIFAR-10 examples were “forgettable.”

3) In the experiment presented in Figure 4b, it is difficult to tell whether the never forgotten set suffers less degradation in the third training regime because the examples were never forgotten or because the model had twice has much prior experience. Please include a control where the order is flipped (e.g., forgotten, never forgotten, forgotten in addition to the included never forgotten, forgotten, never forgotten order currently present).

4) The visual inspection of forgettable and unforgettable examples in Figure 2 is extremely anecdotal, and moreover, do not even appear to clearly support the claims made in the paper.

Minor points:

1) In the discussion of previous studies which attempted to assess the importance of particular examples to classification decisions, a citation to [1] should be added.

2) The point regarding similarity across seeds is absolutely critical (especially wrt major comment 1) , and should be included earlier in the paper and more prominently.

3) The histograms in Figure 1 are misleading in the cropped state. While I appreciate that the authors included the full histogram in the supplement, these full histograms should be included in the main figure as well, perhaps as an inset.

4) The inclusion of a space after the commas in numbers (e.g., 50, 245) is quite confusing, especially when multiple numbers are listed as in the first line on page 4.

[1] Koh, Pang Wei and Percy Liang. “Understanding Black-box Predictions via Influence Functions.” ICML (2017).

---

> ### Author Response · Authors · 2018-11-14
> **Additional Experiments and Clarifications**
>
> Thanks for your detailed review. We tried to improve the paper according to your comments:
>
> -- Major points:
>
> 1) We do acknowledge the importance of considering the possibility of forgetting occurring by chance, suggesting the need for confidence bounds on the number of forgetting events. Before addressing it with additional experiments, we wish to point out that the paper in its current form suggests that it is highly unlikely for the ordering produced by the metric to be the by-product of another unrelated random cause:
>
> 1/ The correlation between the ordering obtained from two sets of 5 random seeds is 97.6%. We will highlight this fact more prominently in the paper (according to your minor point 2).
> 2/ Removing unforgettable examples has a stronger effect than removing randomly chosen examples, suggesting that the vast majority of removed examples with low forgetting events are not picked due to some unrelated random phenomenon.
>
> We followed your interesting suggestion and applied random steps to collect chance forgetting events on CIFAR-10. The results are shown in Appendix 11 of the updated paper. We report the histogram of ``chance forgetting events (please, see text in the paper for more details) averaged over 5 seeds. This gives an idea of the chance forgetting rate across examples. In this setting, examples are being forgotten “by chance” at most twice and most of the time less than once. We are happy to include parts of that section in the main text if it answers your concerns, as we believe it makes the paper stronger.
> We also ran the original experiment on 100 seeds to devise 95% confidence bounds on the average (over 5 seeds) number of forgetting events per example (see Appendix 12). The confidence interval of the least forgotten examples is tight, confirming that examples with a small number of forgetting events can be ranked confidently.
>
> 2) We trained on all datasets for the same number of epochs (200) to study the number of forgetting events. We’ll clarify this in the paper.
>
> 3) Not including the figure with the opposite alternating sequence of tasks was an oversight (we intended to include it in the supplementary). We have now included it in the main paper in Figure 4 (right). Note that the “never forgotten” set continues to suffer from less degradation when training on the “forgotten at least once” set.
>
> 4) We have updated Figure 2 to include a forgettable and unforgettable example from each class, and have included 12 more examples per class in the supplementary (Figure 14). Our main claim is that the unforgettable examples are supported by other examples in the training set, and thus can be removed without impacting generalization. The visualization shows that the unforgettable examples indeed are prototypical of their class (e.g. unobstructed full view of the entire object, commonly observed background), especially when compared to the forgettable examples, which contain more peculiar features (e.g. obstructed view of object or only parts of the object, uncommon color or context).
>
> -- Minor points
>
> 1) We thank the reviewer for pointing us to this work and have included it in the discussion (Section 2 / Paragraph 1)
> 2) We have moved this discussion to Section 4 where we mention experimental results and mentioned the finding at the end of the Introduction.
> 3) We have updated Figure 1 to include the full histograms.
> 4) We’ve updated all numbers to improve readability.

---

> > ### Comment · AnonReviewer1 · 2018-11-16
> > **Response to Authors**
> >
> > Thank you for your response and the additional experiments provided. Please find my comments below:
> >
> > 1) Both of the additional experiments (Appendices 11 and 12) are quite nice and provide clear evidence that the results observed are not merely due to chance forgetting. For Figure 12, please include a comparison to the histogram of forgetting events under true gradient steps as well. In addition, I could not find discussion of chance forgetting in the manuscript itself. Please include several sentences discussing both of these experiments in the main text (it's fine to leave the figures and details in the appendix).
> >
> > 2) Thank you for the clarification.
> >
> > 3) Thank you for including the additional ordering in Figure 4. While these experiments definitely show that the degradation in section 2 is greater for the forgotten set than the never forgotten set, it's interesting that the forgotten set is relatively stable for the first half of c.3, such that the difference between c.3 and b.3 is only present between epochs 50 and 60. I wonder if this is simply due to chance in the training run. It would be helpful to redo this experiment once more with multiple runs and error bars to assess whether this is real or simply an artifact.
> >
> > 4) Thanks for including additional examples in the supplemental figure. Just to clarify, were these examples chosen randomly or hand-selected?
> >
> > In light of the updated results, I have increased my score to a 6. Should the authors include a new version of Figure 4 with multiple runs and address the other post-rebuttal comments, I would be happy to further increase my score.

---

> > > ### Author Response · Authors · 2018-11-19
> > > **Updated Figures and Descriptions**
> > >
> > > Thank you for the response and for the additional comments. Please find our responses below:
> > >
> > > 1) We’ve included the histogram of forgetting events under the true gradient steps in Figure 12 in the updated Appendix 11. We also included a discussion about confidence bounds in the paragraph “Stability across seeds” in Section 4 and we created a new paragraph “Forgetting by chance” to discuss the new results.
> > >
> > > 3) We’ve updated Figure 4 with the mean and standard errors over 5 runs of the experiments. In each run, we randomly sample the ‘never forgotten’ set and the ‘forgotten at least once’ set from all of the examples of their respective kind. The initial stability of the forgotten set in the first half of c.3 is reproducible. This is an interesting observation that we plan to investigate in the future.
> > >
> > > 4) The examples in the supplemental figure are the least and most forgotten examples of each class, when all examples are sorted by number of forgetting events (ties are broken randomly). We clarified this in Appendix 13 in the updated paper.

---

> > > > ### Comment · AnonReviewer1 · 2018-11-19
> > > > **Second Response to Authors**
> > > >
> > > > Thank you for providing the additional experiments and updating the text. The new section on "Forgetting by chance" is very nice and the multiple runs for Figure 4 make the point much more convincingly.
> > > >
> > > > Overall, the paper has improved dramatically since the initial submission, and I appreciate the authors' effort to provide additional controls to clarify and provide additional substantiation for the claims made in the paper. The observations in this work are significant and novel, and as such, I am raising my score to an 8, and clearly recommend acceptance to ICLR.

---

> > > > > ### Author Response · Authors · 2018-11-21
> > > > > **Acknowledging Remarks**
> > > > >
> > > > > Thank you for all of your important remarks -- they have substantially contributed to improving the paper and we will make sure to acknowledge it in the final version of the paper, if accepted.

---

> ### Public Comment · ~Xinshao_Wang1 · 2019-06-04
> **The measurement of forgetting itself**
>
> Dear authors, it is a great work and I am interested in it a lot.
>
> As one offical reviewer mentioned, I am also concerned about the the measurement of forgetting itself. ''Simply due to chance, some examples will be correctly labeled at some point in training, which makes it difficult to distinguish whether a “forgotten” example was actually ever learned in the first place. "
> I suppose many factors cause this phenomenon happening: the random sampling of SGD, the batch size, the learning rate, initialisation, etc.
>
> I noticed that the paramters' update order has been studied and added.
> Could you please share more information about other factors, e.g., batch size, learning rate, or initialised by pretrained models etc.
>
>
> I am looking forward to your sharing. Thanks very much.

---

### Official Review · AnonReviewer3 · 2018-11-15
**Extra Review: Excellent paper which thoroughly explores a very interesting question**

**Rating:** 9
**Confidence:** 5

**Review:**

This is an excellent analysis paper of a very interesting phenomenon in deep neural networks.

Quality, Clarity, Originality:
As far as I know, the paper explores a very relevant and original question -- studying how the learning process of different examples in the dataset varies. In particular, the authors study whether some examples are harder to learn than others (examples that are forgotten and relearned multiple times through learning.) We can imagine that such examples are "support vectors" for neural networks, helping define the decision boundary.

The paper is very clear and the experiments are of very high quality. I particularly appreciated the effort of the authors to use architectures that achieve close to SOTA on all datasets to ensure conclusions are valid in this setting. I also thought the multiple repetitions and analysing rank correlation over different random seeds was a good additional test.

Significance
This paper has some very interesting and significant takeaways.
Some of the other experiments I thought were particularly insightful were the effect  on test error of removing examples that aren't forgotten to examples that are forgotten more. In summary, the "harder" examples are more crucial to define the right decision boundaries. I also liked the experiment with noisy labels, showing that this results in networks forgetting faster.

My one suggestion would be to try this experiment with noisy *data* instead of noisy labels, as we are especially curious about the effect of the data (as opposed to a different labelling task.)

I encourage the authors to followup with a larger scaled version of their experiments. It's possible that for a harder task like Imagenet, a combination of "easy" and "hard" examples might be needed to enable learning and define good decision boundaries.

I argue strongly for this paper to be accepted to ICLR, I think it will be of great interest to the community.

---

> ### Author Response · Authors · 2018-11-21
> **Additional Experiments Introducing Pixel Noise**
>
> Thank you for your review and suggestions.
>
> We performed two additional experiments in CIFAR-10 and have presented the results in the updated supplementary. We are happy to include any parts that the reviewer finds helpful in the main paper.
>
> 1. We corrupt all training images with additive Gaussian noise with mean 0 and increasing standard deviation (std 0.5, 1, 2, 10), and track the forgetting events during training as usual. Note that we add the noise after a channel-wise standard normalization step of the training images (zero mean, unit variance). Therefore, noise with standard deviation of 2 has twice the standard deviation of the unperturbed training data.
>
> We present the results in Figure 11 in Appendix 10.  We observe that adding increasing amount of noise decreases the amount of unforgettable examples and increases the amount of examples in the second mode of the forgetting distribution.
>
> 2. We follow the label noise experiments presented in Figure 3, and augment only random 20% of the training data with additive Gaussian noise (mean 0, std 10). We present the results of comparing the forgetting distribution of the 20% of examples before and after pixel noise was added in Figure 12 (Left) in Appendix 10. We observe that the forgetting distribution under pixel noise resembles the one under label noise. It is a very interesting observation that we plan to investigate in the future.
>
> We agree that it is important to follow-up with a dataset like Imagenet and will pursue this direction in our future work.

---

### Meta-Review · Area_Chair1 · 2018-12-12

**Confidence:** 4
**Recommendation:** Accept (Poster)

**Metareview:**

This paper is an analysis of the phenomenon of example forgetting in deep neural net training. The empirical study is the first of its kind and features convincing experiments with architectures that achieve near state-of-the-art results. It shows that a portion of the training set can be seen as support examples. The reviewers noted weaknesses such as in the measurement of the forgetting itself and the training regiment. However, they agreed that their concerns we addressed by the rebuttal. They also noted that the paper is not forthcoming with insights, but found enough value in the systematic empirical study it provides.